# Factors Included in T1DM Continuing Education for Korean School Nurses: A Systematic Review

**DOI:** 10.3390/ijerph18041620

**Published:** 2021-02-08

**Authors:** Eun-Mi Beak, Yeon-Ha Kim

**Affiliations:** 1Department of Preventive Medicine, College of Medicine, Catholic University of Korea, Seoul 06591, Korea; hanel2004@naver.com; 2Department of Nursing, Korea National University of Transportation, Chungbuk 27909, Korea

**Keywords:** systematic review, school nursing, diabetes mellitus, type 1, education, continuing

## Abstract

(1) Background: The aim of this systematic review was to identify key factors for inclusion in continuing education for Korean school nurses to improve their competence in managing students with type 1 diabetes mellitus (T1DM). (2) Methods: This systematic review was conducted according to the Preferred Reporting Items for Systematic Reviews and Meta-Analyses guidelines. (3) Results: Twelve studies were included in this systematic literature review. The factors identified for inclusion in continuing education on Type 1 diabetes mellitus included 6 competencies. These were strengthening competence in managing students with Type 1 diabetes mellitus, facilitating networking with experts and peers, the perspective of the school nurse as a leader, use of a type 1 diabetes mellitus-specific evidence-based standardized approach of care, supporting self-management to promote healthy learners, and communication and collaboration between key stakeholders. Identified barriers to accessing continuing education on type 1 diabetes mellitus were work demands, difficulty taking time off during the school year, and limited support from administrators. (4) Conclusions: Based on the findings of this study, online or e-learning continuing education on type 1 diabetes mellitus must be developed for school nurses who manage students with this condition.

## 1. Introduction

### 1.1. Introduction

Type 1 diabetes mellitus (T1DM) is one of the most common chronic childhood illnesses. In individuals with T1DM, the pancreas does not produce the necessary insulin to properly maintain blood glucose [1,2]. T1DM requires maintenance of a target glucose level, reliance on insulin infusion, and other health management practices. Among children with T1DM, parental involvement and school support significantly impact overall diabetes control. School nurses are becoming increasingly involved in this process and play an essential role in helping students cope with T1DM [3,4].

The role of the school nurse in managing students with T1DM is to improve self-management, allowing these students to lead a healthy school life [5]. Thus, it is vital for school nurses to possess knowledge and skills associated with managing students with T1DM and to provide emergency nursing care when needed. However, school nurses have expressed fears resulting from a perceived lack of competence [3,6,7,8,9].

Given the complexity of caring for students with T1DM and the frequent changes in treatment protocols, continuing education (CE) is critical for school nurses to develop their professional expertise and to update their knowledge, skills, and attitudes [7]. School nurses should engage in lifelong learning and practice through CE, which must be a systematic professional learning experience. The unprecedented crisis of the global COVID-19 pandemic has led to a massive shutdown of face-to-face activities, and interruptions to CE can have long-term implications. Alternative learning pathways must be sought to limit the interruption to education as best as possible. To optimize the development of T1DM CE in this contactless era, it is important to review studies on T1DM management by school nurses and to find key points to address, such as their needs, competence levels, and potential barriers. However, little attention has been devoted to the systematic re-view of T1DM management among school nurses to establish basic data for CE development. Most of the previous literature reviews of T1DM and school nurses did not involve systematic processes [10] or conduct a quality appraisal [11,12,13]. Only one study was found to have a systematic review methodology and to evaluate quality, but the purpose of that review was to synthesize the views of children, parents, and professionals in self-care and management [14]. Currently, no study has provided a systematic analysis of evidence regarding factors to incorporate into CE for school nurses for the management of students with T1DM.

The aim of this systematic review was to synthesize findings from empirical studies and identify related factors for inclusion in CE for Korean school nurses to improve their competence in managing students with this chronic condition.

### 1.2. Conceptual Framework

Importantly, CE in this context should be developed based on a theoretical framework to identify key competencies to improve among school nurses [15,16,17]. The theoretical model used in the research incorporated Benner’s novice-to-expert theory [18] and the healthy learner model [9]. Based on the novice-to-expert theory, the roles of school nurses have been suggested to be helping, teaching, and coaching, diagnostic monitoring, effectively managing rapidly changing situations, administering and monitoring therapeutic interventions and regimens, ensuring quality health care practices, and organizational and work role competencies [18]. According to the healthy learner model, the competencies of school nurses in optimizing the health and supporting the academic success of children with chronic conditions have been suggested to be leadership, evidence-based practice, capacity building, serving as a dedicated resource for information on chronic diseases, promoting healthy learners, engaging in partnerships with families, and engaging in partnerships with health care providers [9]. Based on these two theories, we identified related factors for inclusion in CE for Korean school nurses to improve their competence in managing students with T1DM.

## 2. Materials and Methods

### 2.1. Study Design

The aim of this systematic review was to identify key factors for inclusion in CE for Korean school nurses to improve their competence in managing students with T1DM.

### 2.2. Inclusion Criteria and Exclusion Criteria

The inclusion criteria for studies to be analyzed in this review were determined based on the Population, Intervention, Comparison, and Outcomes (PICO) framework proposed by the Cochrane Collaboration Group for systematic literature reviews [19]. PICO elements were adopted for the systematic literature review via discussion between the researchers. As the purpose of the study was not to evaluate efficacy, search words were selected that focused on the study group (P) and intervention (I) without any restriction on the comparison group (C) or the outcome variable (O). The subjects (P) of the current study were studies involving school nurses, and the intervention (I) was literature describing “diabetes mellitus” and “diabetes mellitus, type 1”.

#### 2.2.1. Inclusion Criteria

(1)Studies involving school nurses(2)Non-randomized studies(3)Studies referring to school nurses among school staff(4)Studies mentioning the management of students with diabetes in elementary, middle, and high schools(5)Studies involving both type 1 and type 2 diabetes, as long as type 1 diabetes was included

#### 2.2.2. Exclusion Criteria

(1)Brief updates and letters to the editor(2)Documents for which the full text could not be obtained(3)Studies not published in English or Korean(4)Position statements from associations(5)Studies in which schoolteachers, parents, or students were the main subjects(6)Studies not related to T1DM management by school nurses(7)Gray literature (academic conference presentation materials and abstracts without full texts, theses/dissertations, editorials, etc.)

### 2.3. Search Methods

In this study, based on the flow chart for a systematic literature review [20] according to the Preferred Reporting Items for Systematic Reviews and Meta-Analysis guidelines, the studies to be analyzed were selected through identification, screening, eligibility assessment, and inclusion. A literature search was conducted using electronic databases and other literature search methods. The databases used were MEDLINE (Ovid), Embase (Ovid), Cochrane Library (Wiley), and CINAHL (EBSCO) for foreign studies and the Korea Education and Research Information Sharing Service (RISS) the Korean Studies Information Service System (KISS), the academic information portal DBpia, and the National Digital Science Digital Library (NDSL) for domestic studies.

The keyword selected for the population was “school nurses”, the keywords for intervention were “diabetes” and “student” for interventions 1 and 2 (I-1 and I-2), respectively, and the search formula consisted of a P AND (I-1 AND I-2) strategy. The search terms selected for the population were “school health nursing”, “health educators”, “health education”, “school health services”, and “school nursing”. The search terms selected for I-1 were “diabetes mellitus”, “diabetes mellitus, type 1”, and “diabetes mellitus, type 2”, and those selected for I-2 were “students”, “child”, and “adolescent”.

The MeSH (Medical Subject Headings) controlled vocabulary was used on MEDLINE and Cochrane Library, and the Emtree controlled vocabulary was used on Embase. After the controlled vocabulary was selected, natural language was added to determine the search terms, and Boolean operators (AND, OR, and NOT) were used between search terms to generate a search formula. In this study, no additional limiter was used to increase the search sensitivity.

In the Korean language literature, “school nurses” and “diabetes” were searched, and Boolean operators (AND, OR, and NOT) were used between search terms to generate a search formula.

To yield a more accurate search, we consulted six information search experts and conducted the main literature search on 21 March 2020. Regarding other literature search methods, we visited websites directly and searched manually, reviewing each document.

### 2.4. Literature Selection

A total of 25,301 articles were identified through the literature search, and 11 Korean studies were identified. The titles and content of a total of 24,971 articles (excluding 330 duplicate articles) were reviewed. Based on this review, 24,953 articles were excluded, and 18 articles were selected for inclusion. After reviewing the abstracts of the 18 articles, 6 were excluded, and the full texts of the remaining 12 articles were reviewed. The 12 articles that met the inclusion criteria were selected for the systematic review and quality evaluation. To increase the degree of agreement, the entire process of literature selection was conducted after the two researchers reached a consensus on that process through a preliminary meeting. In the primary exclusion process, the title and abstract were reviewed to determine whether the study satisfied the literature selection criteria. When this was difficult to ascertain based solely on the title and abstract, the study was included and was then subjected to a professional review in the secondary exclusion process for a final decision (Figure 1).

### 2.5. Quality Evaluation of Studies Included in the Systematic Review

The quality evaluation of the 12 papers selected for the systematic literature review was independently performed by two researchers. Any disagreement in the evaluation of each paper between the researchers was supposed to be resolved by a third-party nursing professor, but no case necessitated a third-party review. A critical review of the literature was conducted using the Risk of Bias in Non-randomized Studies quality assessment tool [21]. The quality evaluation results were entered into RevMan and analyzed for each section.

### 2.6. Data Analysis

All documents identified through the systematic literature review were entered into and managed within a literature management program (EndNote X9, Clarivate Analytics, Philadelphia, PA, USA). Systematic confirmation, synthesis, statistical merging, and outcome analysis of the selected studies were conducted according to the Cochrane guidelines [19]. To collect the characteristics of the 12 included studies and the key findings regarding T1DM CE for school nurses, two researchers developed a coding sheet, and the content of the analysis was coded and organized. The coding sheet included the following items: author, year, country, objectives, research design, study size, analytical tool, and key findings. The coding sheet was filled out independently by one researcher and then verified jointly with the other researcher to prevent typographic errors.

## 3. Results

### 3.1. Characteristics of the Studies Included in the Systematic Review

This systematic literature review included 12 studies, all of which were original articles. Six (50.0%) articles were published between 2000 and 2009, and six (50.0%) were published between 2010 and 2019. The majority of the studies were conducted in the United States (*n* = 10), one in Taiwan, and one in Korea. Two of the 12 studies were qualitative, while the remaining 10 studies were quantitative.

Of the 10 quantitative studies, four involved a survey design, three involved a program evaluation, one was a cohort study, and two were conducted before and after interventions. In four studies, a theoretical framework was used for the development of interventional programs. The majority of studies addressed both type 1 and type 2 diabetes in school settings (Table 1).

### 3.2. Description of the Studies Included in the Systematic Review

The authors identified themes of related factors for inclusion in CE for Korean school nurses to improve competency in managing students with T1DM. These are described in Table 1. Through an analysis by consensus of 17 studies, the following competencies were identified: strengthening competence in managing students with T1DM [6,7,8,9,15,16,17,22,23,24,25], facilitating networking with experts and peers [6,7,17], the perspective of the school nurse as a leader [15,17,22,23,24,25,26], the use of a T1DM-specific evidence-based standardized approach of care [7,8,9,15,17,22,23,24,25,26], supporting self-management to promote healthy learners [8,15,17], and communication and collaboration between key stakeholders [6,7,9,15,22,25,26].

Barriers to the administration of CE in this context were identified. Job demands including factors such as a demanding work schedule [16,25] and difficulty taking time off during the school year [7] were pointed out as contributors to difficulties in participating in CE. Time constraints [7,16,25], financial cost [7,16], the necessity for travel [16], difficulty taking time off during the school year [7], lack of access to educational training [16,25], difficulty using computer technology [7], and a lack of regularly updated educational resources [25] were described as barriers. Low opinions of school nurse professionalism [25], as well as the hesitation of administrators to pay course fees [8] and provide paid leave [8], were also noted as barriers to participation in CE.

### 3.3. Quality Appraisal of Studies Included in the Systematic Review

The results of the RevMan quality evaluation of the 12 selected studies are as follows: of the 12 studies analyzed using the Risk of Bias in Non-Randomized Studies tool [21], all 12 were classified as having a low risk of selection bias with regard to the selection of participants, while 10 were categorized as unclear risk and two as low risk with regard to the failure to consider confounding variables. Regarding the measurement of exposure, eight studies were rated as having an unclear risk of bias, one as high risk, and three as low risk. With regard to the blinding of the outcome assessments, four studies were considered to have an unclear risk of bias and eight to be high risk, and no study had a low risk of bias. Regarding the presence of incomplete outcome data, the risk of attrition bias was found to be low for all papers. With respect to selective outcome reporting, all studies were classified as having a low risk of bias, as the protocols and plan contents were con-firmed. However, we included all of these studies in this systematic review for the purpose of reviewing the key findings and presenting comprehensive study results (Figure 2).

## 4. Discussion

Many students struggle with T1DM in school settings, and school nurses are thus required to react effectively to help students and their families manage this condition. Nevertheless, these nurses find themselves falling short with regard to competence; therefore, the provision of CE on T1DM is important to strengthen this competence. In this study, we identified six competencies required for inclusion in a T1DM CE program for Korean school nurses based on the healthy learner model [27] and Benner’s novice-to-expert theory [18] to improve the management of students with T1DM. These competencies included strengthening competence in managing these students, facilitating networking with experts and peers, the perspective of the school nurse as a leader, use of a T1DM-specific evidence-based standardized approach of care, supporting self-management to promote healthy learners, and communication and collaboration between key stakeholders.

Strengthening competence in managing students with T1DM was found to be an important competency [6,7,8,9,15,16,17,22,23,24,25]. Learning style preferences differed among school nurses. T1DM CE was effective via online [7] and e-learning [16] modalities; however, some school nurses preferred a class setting with face-to-face contact [7,8]. These researchers emphasized the importance of providing a discussion board at the end of online CE on T1DM and allowing sufficient time for open ended questions to provide additional information [6,7]. Nonetheless, it is also important to seek strategies for active learning, interact with the material to enhance comprehension, and give feedback to the learner [6,16]. However, as COVID-19 has led to global restrictions on face-to-face activity, education has changed dramatically, with the distinctive rise of online education and e-learning. Therefore, online or e-learning T1DM CE programs for school nurses are currently preferred. Many studies have also pointed out that school nurses have encountered shortages of knowledge and skills regarding advanced diabetes treatment and technologies, which others have identified as an essential competency of school nurses in managing students with T1DM [6,7,8,9,15,16,17,22,24,25]. Their education needs were related to caring for students with insulin pumps or glucagon injections rather than knowledge of the pathology of diabetes; thus, the use of advanced diabetes techniques such as insulin pumps [6,7,8,9,15,16,22,24,25], the use of diabetes equipment such as glucose monitors [7,9,16,17,22], and the recognition and treatment of hypoglycemia in emergency situations [6,15,16,23,24,25] should be emphasized in T1DM CE. Although school nurses preferred face-to-face contact for the enhancement of diabetes skills [7,8], it is not easy to design and implement a T1DM CE program involving face-to-face contact due to the current pandemic; therefore, virtual reality simulation (VSim) has emerged as an alternative educational method [28]. As previous studies have pointed out, school nurses need regular practice with sufficient time to improve diabetes-related skills [6,7]; the VSim method can solve this problem, as it is flexible and highly accessible, with a remote function. However, none of these studies explored this training method, limiting the capacity of this review to discuss it.

Facilitating networking with experts and peers also appeared to be an important requirement in T1DM CE for school nurses [6,7,17,22]. Networking with colleagues and local professionals through CE helps school nurses to identify valuable resources and establish useful contacts in the community [6,17,22]. A previous study suggested arranging time in a T1DM CE program for school nurses to share ideas and experiences with peers and to connect for one-on-one coaching and mentoring [7].

The perspective of the school nurse as a leader was found to be another required competency [10,15,17,22,23,24,25,26]. School nurses must be aware of laws requiring policies and protocols for the care of students with diabetes while in school [15,23,24]. Moreover, they need to participate in the creation of diabetes-related school policies [23]. In Korea, school nurses are legally allowed to provide emergency care to students with T1DM in cases of hypoglycemic shock (School Health Act, Article 15-2). As they can be responsible for providing emergency nursing care in life threatening situations, school nurses find themselves experiencing role confusion due to an absence of legal protection from medical accidents [15]. Therefore, CE regarding T1DM-specific law in school settings is needed for school nurses so that they can practice nursing care according to proper legal and ethical principles [29].

The findings reinforce the appropriateness and importance of a T1DM-specific evidence-based standardized approach of care in CE programs [6,7,8,9,15,17,22,23,24,25,26]. In previous studies, educational workshops for school nurses were developed based on key components and evidence-based T1DM management [7,9,16,22]. The need to use DM-specific evidence-based clinical guidelines to ensure effective T1DM management has been emphasized repeatedly in the literature [6,7,8,15,17,23,24]. School nurses have acknowledged the inadequacy of their knowledge and skills due to consistent advancements in DM technology, yet these nurses often lack access to updated resources, which lowers their confidence with regard to diabetes management [17]. Thus, it is important to develop updated educational materials and guidelines when delivering T1DM CE. Providing educational materials and guidelines and allowing school nurses to review updated information on diabetes and insulin pumps prior to the training could make CE more effective [7]. The importance of an optimal diabetes management plan is also related to the T1DM-specific evidence-based standardized approach of care [7,9,22,23,24,25,26]. Given that the preparation of individual health plans (IHPs) appears to be vital to diabetes management, competence in developing them must be increased. Previous studies described using the results of the Diabetes Control and Complications Trial in developing IHPs [24]. IHPs include nursing diagnoses, student outcome goals, outcomes, indicators, interventions, and delegation, and emergency health plans include glucose monitoring, insulin dosage, snacks, exercise, and protocols for hypoglycemia and hyperglycemia. These plans must be evaluated yearly and adjusted based on the student’s developmental stage, emotional maturity, and behavior [9,22,23,25,26]. School nurses are often inadequately trained, and many are not able to help students manage their T1DM based on IHPs. Therefore, developing standardized Korean versions of IHPs and training on this topic in T1DM CE programs is vital and urgent.

School nurses have been shown to require training in T1DM CE to support student self-management of T1DM in order for these students to be healthy learners [8,17]. To provide diabetes education on self-management, school nurses must increase their competence in the assessment of pediatric development and behavior so that they can coordinate care for students [8,17]. A previous study reported that school nurses exhibit lower confidence in providing diabetes education when the school has a lack of access to resources or diabetes equipment, and only nine (5.5%) out of 70 school nurses were found to have prepared a diabetes lesson plan to educate students with diabetes [17]. As the self-management of T1DM is developed in a learning process, school nurses need training on how to provide individualized education for each student based on a diabetes lesson plan that recognizes the student’s development stage and maturity [17].

Communication and collaboration between key stakeholders were another required competency [6,9,15,22,23,25,26]. The findings show a broad consensus on the need for school nurses to be competent at creating teamwork with students [15,22,25,26], parents [15,22,23,25], school personnel [15,22,25], health providers [15,22,25], and any others who interact with students. Regular communication with stakeholders regarding the needs of students with T1DM and the exchange of information were found to be beneficial [26]. Although school nurses need collaborative connections with students’ physicians, many practical limitations exist, since physicians are usually busy and difficult to reach [25]. However, school nurses experience a lack of adequate communication training, and many are unable to promote teamwork [25]. In this contactless era, we must train school nurses on how to communicate and collaborate with students, parents, and school personnel using online methods.

Barriers to the participation of school nurses in T1DM-related CE have emerged. In Korea, any elementary school with at least 18 classes, or any middle or high school with at least nine classes, should employ one school nurse (Enforcement Degree of the School Health Act, article 23). A school nurse to student ratio of 1:750 is recommended by the U.S. Department of Health and Human Services, and this number of students is reduced in situations involving students with complex health needs (US. 2012); however, the ratio is 1:1000 in Korea (Korean Educational Statistics Service, 2019). As such, school nurses in Korea and other countries are exposed to greater numbers of students and face particularly intense job demands [16,25]. Thus, they need backup coverage at school [7,16,25] or time for them to take leave during the school year [7]. Therefore, it is necessary to establish policies to employ extra school nurses in schools with more than a certain number of students to reduce workload and allow for backup coverage in cases of emergency. Cost and time constraints [7,16,25] and the necessity for travel [16], which were also barriers for school nurse participation in T1DM CE, can be resolved by promoting online T1DM CE; however, it is necessary to provide a detailed guide on how to access these programs on a computer [7]. Support from administrators was found to be an important factor in school nurse participation in T1DM CE [8,25]; therefore, interventions to change the perceptions of administrators regarding school nurse professionalism must be conducted [25].

This study had several limitations. First, the sample sizes of the included studies were relatively small. Hence, caution should be used when generalizing these findings. Second, the comparison and outcome aspects of the PICO model, such as specific intervention effects, were not included in the search terms. This omission reflected our aim of determining related factors for inclusion in T1DM-related CE for Korean school nurses rather than elucidating the effects of this CE; however, future studies are recommended. Third, we refrained from conducting a meta-analysis. This was done intentionally, to avoid the so-called “garbage in, garbage out” problem; regardless, a further meta-analysis is needed. In addition, only a few studies included in this review were determined to be of high quality.

## 5. Conclusions

It is important to provide CE on T1DM to school nurses to strengthen their competence in managing students with this condition. In this study, we identified six competencies required for inclusion in a T1DM CE program for Korean school nurses based on the healthy learner model [27] and Benner’s novice-to-expert theory [18]. These competencies included strengthening competence in managing these students, facilitating networking with experts and peers, the perspective of the school nurse as a leader, using a T1DM-specific evidence-based standardized approach of care, supporting self-management to promote healthy learners, and communication and collaboration between key stakeholders. Currently, CE on T1DM is facing challenges due to COVID-19 and the associated need to change education methods; in this context, online or e-learning T1DM CE could be superior to the CE methods currently used, as it can be tailored to cover identified knowledge gaps among various groups [16]. These methods can provide effective education in a time- and cost-sensitive way and can easily address barriers such as high work demands, the inability to take leave during the school year, and administrator support for school nurse participation in T1DM CE. Online or e-learning-based T1DM CE for school nurses must be developed to train diabetes-related skills with VSim, facilitate networking with experts and peers, include discussion boards, and promote communication with stakeholders. The content must include the assessment of pediatric developmental and behavioral stage, T1DM-specific law in school settings, and up-to-date information on T1DM treatment. T1DM-specific evidence-based clinical guidelines and standardized Korean versions of IHPs need to be developed and taught in T1DM CE programs. Further study is needed to verify the effects of T1DM CE programs for school nurses.

## Figures and Tables

**Figure 1 ijerph-18-01620-f001:**
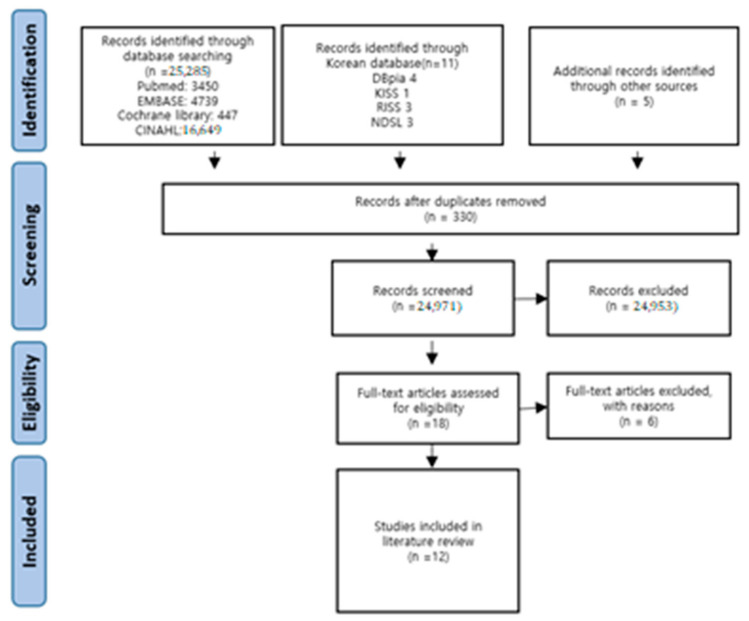
Flow chart of the study selection process.

**Figure 2 ijerph-18-01620-f002:**
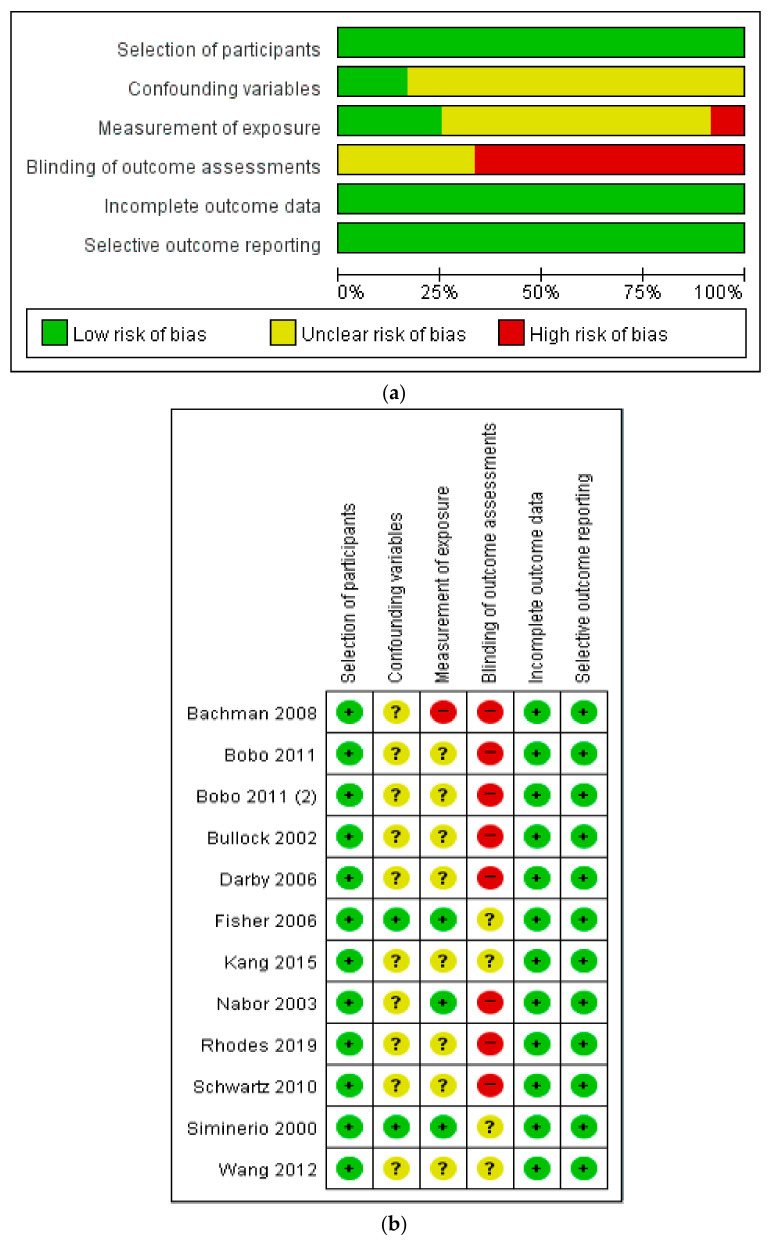
(**a**). Risk of bias graph [20]. (**b**). Risk of bias summary. Reviewing authors’ judgments about each risk of bias item presented as percentages across all included studies.

**Table 1 ijerph-18-01620-t001:** Description of the studies included in the systematic review.

Author/ Year/Country	Objective(s)	Research Design	Study Size/ Analytical Tool	Key Findings
Bachman et al. (2008) [7] US	To develop and evaluate an online CE program to educate school nurses in how to manage the care of children with diabetes in school using the current practice principles	Program evaluation	Fifteen school nurses/online diabetes management CE program	▪ Program: An overview and update on diabetes management, insulin pump use, and the role of the school nurse ▪ A manual should be developed ▪ Peer sharing of ideas and experiences is valuable ▪ Nurses need to apply information to care plans ▪ Online CE on current practices for care andpump technology enhances school nurses’ abilityto manage children with diabetes ▪ Barriers: (a) difficulty using computer technology, (b) time constraints and family responsibilities, (c) limited finances, (d) a lack of backup coverage at school, and (e) difficulty getting away during the school year
Bobo et al. (2011) [9] US	To describe how the healthy learner model was used to improve practicing school nurses’ ability to provide effective and consistent care to students with diabetes	Program evaluation	Thirty-two to ninety-five school nurses from 2008 to 2009/the Management and Preventing Diabetes and Weight Gain program	▪ Program: Leadership, evidence-based practice, capacity building, nurses as resources for information on chronic diseases, healthy learners, partnership with families, partnership with health care providers ▪ Implementing each student’s individual diabetes medical management plan is important ▪ Nurses need knowledge of blood glucose control, technological advances in insulin delivery, and blood glucose monitoring devices
Bobo et al. (2011) [22] US	To describe how collaboration between key stakeholders resulted in a statewide program in Colorado to meet the needs of students with diabetes and their parents	Program evaluation	Two hundred and forty school nurses/the Colorado Diabetes prevention and Control program/3 years	▪ Program: Promote collaboration with key stakeholders, develop standardized documents (healthcare provider orders document focused on the medical order, independent management plan, emergency care plan, insulin pump monitoring log); the diabetes resource nurse ▪ Nurses need knowledge of insulin pumps, hyperglycemia, and glucagon policies ▪ Determining available resources is required
Bullock et al. (2002) [8] US	To examine how the perceived competence of school nurses who attended a specific CE course differed from that of school nurses who did not attend the course	Cohort study	Five hundred and thirty-seven school nurses (120 who had completed CE and 417 who had not completed CE)/Missouri Department of Health and Senior Services CE programs/35-item questionnaire to measure perceived competence in practice	▪ Program: six CE programs pertaining to diabetes management ▪ Nurses need knowledge of updates to DM-specific evidence-based clinical guidelines, assessment of pediatric development and behavior, and culturally competent coordinated care for children and their families ▪ Nurses need technical skills related to insulin pump function ▪ School nurses who completed face-to-face CE had a higher level of perceived confidence in caring for children with diabetes ▪ Barrier: School administrators are hesitant to pay course fees and provide paid leave
Darby et al. (2006) [6] US	To examine school nurses’ experiences with insulin pump therapy in a school setting	Qualitative study	Eleven school nurses/Research question: nurse perceptions, resources for school nurses, and challenges encountered by nurses	▪ Nurses need to gain hands on experience with insulin pumps ▪ Nurses need updated educational materials and information ▪ Resources from a regional children’s hospital can be used to gain knowledge and hands on experience with pumps ▪ Nurses need training on reacting quickly to assess and treat the symptoms associated with hyperglycemia and hypoglycemia
Fisher (2006) [17] US	To investigate school nurses’ perceptions of self-efficacy in relation to caring for and educating students about diabetes and to identify factors related to self-efficacy in performing these skills	Exploratory survey	Seventy school nurses/the Self-Efficacy Diabetes Education questionnaire	▪ Importance of CE for current practice guidelines ▪ Facilitated networking with colleagues and local professionals, establishing resource contacts, and identifying valuable resources in the community ▪ Lack of access to resources and meter equipment lowers confidence in performing diabetes education ▪ Nurses need a diabetes curriculum (lesson plan) to educate students with diabetes
Kang et al. (2015) [15] Korea	To explore school nurses’ experience, perceived barriers, and education needs related to diabetes management at school	Cross sectional study	One hundred and one school nurses/35-item questionnaire to measure status, barriers, and needs related to diabetes management	▪ Nurses need education on legal responsibility ▪ They also need education on emergency responses (such as glucagon injection and the management of hypoglycemia) ▪ Guidelines should be prepared ▪ Nurses need communication with students, parents, and healthcare workers
Nabors et al. (2003) [26] US	To use survey methods to examine perceptions of barriers to and support for diabetes management during school and after school activities	Survey	One hundred and ten school nurses/survey and open-ended questions	▪ Nurses should train to improve communication with students who have diabetes about their condition ▪ Nurses should also prepare individual care plans
Rhodes et al. (2019) [16] US	To assess the effectiveness of a rapid e-learning module for school nurse professional development in school-based diabetes management	Before and after study	Pretest: 678 school nurses, posttest: 449/rapid e-learning CE/15 multiple choice and five true-false items derived from the learning objectives, information, and resource, with scores ranging from 0 to 20	▪ Program: Learning objects, purpose of DM management, training level of school staff; information and content provided, update and statement, equipment, dietary practices, psychological impacts, type 2 diabetes management techniques, emergency care; synthesis of ideas; question-asking; real world applications, links to free resources from credible medical sources to support the updates ▪ Barriers: (a) demanding work schedule, (b) time intensive professional development opportunities, (c) expense and requirement for travel, (d) lack of access
Rhodes et al. (2010) [23] US	To evaluate the experience of children and adolescents with T1DM in school by surveying children, their parents, and school personnel directly involved in their care	Survey	Eighty children, their parents, and 28 school personnel (95% school nurses)	▪ Staff should develop an individualized diabetes management plan for each child ▪ Staff should also provide sick day assessment information about glucose testing and insulin pumps to parents ▪ Guidelines are required for the care of children with diabetes while in school ▪ Staff should prepare to assist children with hypoglycemia and hyperglycemia and develop emergency management protocols ▪ Awareness of federal laws is important, along with specific policies and protocols for the care of children with diabetes while in school
Siminerio et al. (2000) [24] US	To assess the knowledge levels and needs of school personnel and to determine the effectiveness of a comprehensive diabetes education program that highlights the current trends and associated advanced technologies in the care of children with diabetes	Before and after study	One hundred and fifty-six school nurses and teachers/10-question diabetes knowledge assessment tool -Three additional surveys about needs, experiences, and concerns	▪ Program: information about causes, classification, complications (acute and chronic), care, and cure (the 5 Cs of diabetes) ▪ Diabetes-related information should be provided to school personnel ▪ The recognition and treatment of hypoglycemia must be improved ▪ School staff need awareness of legal protections that apply to children with diabetes ▪ Use information from the DCCT ▪ Train skills related to using rapid acting insulin, pen devices, and insulin pumps
Wang et al. (2013) [25] Taiwan	To obtain a fundamental understanding of school nurses’ lived experiences of caring for students with T1DM	Qualitative study	Five Taiwanese school nurses/In-depth interview, “What is your experience in caring for students with T1DM?” Elementary, junior school	▪ Nurses need to understand developmental stage in the context of peer relations ▪ They also need to advance their knowledge of current diabetes treatment methods and technologies ▪ Increase competence in developing individualized management plans ▪ Require teamwork among nurses, parents, teachers, physicians, peers, and any others who interact with students ▪ Due to practical limitations, emergency care becomes nurses’ priorityBarriers: heavy workload, time constraints, hectic work environment, lack of adequate interdisciplinary training, and school personnel and administrators’ low opinions regarding the professionalism of school nurses

T1DM—type 1 diabetes mellitus; CE—continuing education; DCCT—Diabetes Control and Complications Trial.

## Data Availability

Not applicable.

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
