# Peer review of "Factors Included in T1DM Continuing Education for Korean School Nurses: A Systematic Review"

_ijerph, 2021, doi:10.3390/ijerph18041620_

Round 1

Reviewer 1 Report

The article addresses a very relevant topic as is the competence of school nurses in the care of Type 1 Diabetes Mellitus.

The methodology used for the systematic review is adequate ant it has been carried out in a very rigorous way. However, I believe that more current studies on this topic should be included. The authors should look for studies and systematic reviews published in the last 10 years and specially in the last 5 years.  I think it is critical for this research, to carry out an update of the systematic review.

SUMMARY:

Q12 Results: I suggest to the authors make sentences shorter

MATERIAL AND METHODS

The inclusion and exclusion criteria of the study are very clear. However I think it would be better only to include in the study papers published in the last 10-12 years. It is necessary search again for more current papers.

I send you some examples:

Exploration of School Nurses' Perception of Self-Efficacy in Providing Care and Education to Children with Type 1 Diabetes Mellitus.
Williams LF, Russ M, Perdue BJ.J Natl Black Nurses Assoc. 2019 Dec;30(2):34-37.

Actual and perceived knowledge of type 1 diabetes mellitus among school nurses.
Kobos E, Imiela J, Kryczka T, Szewczyk A, Knoff B.Nurse Educ Today. 2020 Apr;87:104304. doi: 10.1016/j.nedt.2019.104304. Epub 2019 Nov 22.PMID: 32014799

Equipping School Health Personnel for Diabetes Care with a Competency Framework and Pilot Education Program.
Berget C, Nii P, Wyckoff L, Patrick K, Brooks-Russell A, Messer LH.J Sch Health. 2019 Sep;89(9):683-691. doi: 10.1111/josh.12806. Epub 2019 Jun 27.

Diabetes Care in the School and Day Care Setting American Diabetes Association Diabetes Care 2011 Jan; 34(Supplement 1): S70-S74. https://doi.org/10.2337/dc11-S070

LIMITATIONS

Must be explained how could they be avoided and what would be the next steps to take to improve this research

BIBLIOGRAPHY

Some of the studies included were written more than 15 years before. More current and advanced studies should be included in this study

Author Response

Response to Reviewer 1 Comments

I would like to express our sincere gratitude for your thorough consideration and scrutiny over our manuscript. Through the accurate and keen comments made by the reviewer, the critical points at issue in the overall manuscript were discovered and subsequently corrected. After receiving the reviewers’ criticisms, my colleagues and I have extensively revised the manuscript in order to achieve the proper scientific and literary levels required by the reviewer and International Journal of Environmental Research and Public Health.. I hope this revised manuscript will be considered positively and be accepted by and International Journal of Environmental Research and Public Health.

Point 1: SUMMARY Q12 Results: I suggest to the authors make sentences shorter

Response 1: Thank you for your suggestion. We have revised the sentence into;

“Factors Included for continuing education on Type 1 diabetes mellitus were 6 competencies. It were strengthening competence in managing students with Type 1 diabetes mellitus, facilitating networking with experts and peers, the perspective of the school nurse as a leader, use of a Type 1 diabetes mellitus specific evidence-based standardized approach of care, supporting self-management to promote healthy learners, and communication and collaboration between key stakeholders” in L12

Point 2: MATERIAL AND METHODS-I think it would be better only to include in the study papers published in the last 10-12 years. It is necessary search again for more current papers.

Response 2: We thank the reviewer for bringing our attention to these points.

To include current studies on this topic, my colleague and I especially looked for studies published in the last 5 years to carry out an update of the systematic review. Among total of 25,301 articles, studies were excluded with the criteria of studies that does not have full text, were not an original article and rather position statements from the associations, were not related to T1DM, etc. Especially studies were excluded if it were not suitable for quality evaluation(as our study purpose was to provide quality evaluated systemic review).

Some examples which reviewer 1 has suggested were not included because of the following reasons.

Exploration of School Nurses' Perception of Self-Efficacy in Providing Care and Education to Children with Type 1 Diabetes Mellitus. Williams LF, Russ M, Perdue BJ.J Natl Black Nurses Assoc. 2019 Dec;30(2):34-37

-This study was not included because there was no full text. Actual and perceived knowledge of type 1 diabetes mellitus among school nurses.

Kobos E, Imiela J, Kryczka T, Szewczyk A, Knoff B.Nurse Educ Today. 2020 Apr;87:104304. doi: 10.1016/j.nedt.2019.104304. Epub 2019 Nov 22.PMID: 32014799  

-It was not included because the study was opened in 2020 April.

Diabetes Care in the School and Day Care Setting American Diabetes Association Diabetes Care 2011 Jan; 34(Supplement 1): S70-S74. https://doi.org/10.2337/dc11-S070

-This study was more near to an position statement than an original paper therefore was not suitable for quality evaluation.

 Equipping School Health Personnel for Diabetes Care with a Competency Framework and Pilot Education Program. Berget C, Nii P, Wyckoff L, Patrick K, Brooks-Russell A, Messer LH.J Sch Health. 2019 Sep;89(9):683-691. doi: 10.1111/josh.12806. Epub 2019 Jun 27.

-Key word of this study was chronic disease. In our PICO search strategy, population was “school nurses” and intervention were “diabetes’ and “student” therefore, that is why it was not included in our data.

Point 3: Must be explained how could they be avoided and what would be the next steps to take to improve this research

Response 3: Thank you for your suggestion. Do you mean “garbage in, garbage out” problem? We have already described “We refrained from conducting a meta-analysis. This was done intentionally, to avoid the so-called “garbage in, garbage out” problem; regardless, a further me-ta-analysis is needed. “  (L355)

We hope this sentence satisfy your comment.

Point 4: Some of the studies included were written more than 15 years before. More current and advanced studies should be included in this study

 Response 4: : Thank you for your suggestion. Two references were more than 15 years. We have revised reference 1. “Chiang, J.L.; Maahs, D.M.; Garvey, K.; Hood, .KK.; Laffel, L.M., Weinzimer, S.A.; Wolfsdorf, J.I.; Schatz, D. Type 1 Diabetes in Children and Adolescents: A Position Statement by the American Diabetes Association. Diabetes Care. 2018, 41(9), 2026-2044, doi: 10.2337/dci18-0023.”   (L396)

And reference 10 were left behind. Because there were a few previous literature reviews of T1DM and school nurses and we needed to compare the difference of our study in the introduction.  (L416)

Reviewer 2 Report

Thank you for this important contribution, which I enjoyed reviewing. I have no further comments to make.

Author Response

I would like to express our sincere gratitude for your thorough consideration and scrutiny over our manuscript.

Reviewer 3 Report

This is a well designed and executed review. All the necessary aspects of a rigorous systematic review are present - it follows PRISMA principles; uses PICO to derive the review question; summarises the retrieved study and assesses bias in the studies.

In a few places in the manuscript - including in the abstract - there is some random hyphenation (eg 'This systemat-ic review') that needed to be addressed before publication.

I would only make one suggestion and that is - to complete the PRISMA process - you could complete a PRISMA checklist, as far as possible, and then add to the submission as a 3rd piece of supplementary information.

Author Response

I would like to express our sincere gratitude for your thorough consideration and scrutiny over our manuscript.  I hope this revised manuscript will be considered positively and be accepted by and International Journal of Environmental Research and Public Health.

Our responses to the reviewer’s comments are as follows:

Point 1:  in the abstract - there is some random hyphenation (eg 'This systemat-ic review')

Response 1: Thank you for your suggestion. We have revised the sentence. (L10)

Point 2: complete a PRISMA checklist, as a 3rd piece of supplementary information.

Response 2: Thank you for your suggestion. We have  uploaded PRISMA checklist as a file

Round 2

Reviewer 1 Report

The article issue is very  interesting.  

I encourage you follow this research 

Reviewer 3 Report

I was very impressed with this manuscript first time round and I remain happy with it.